# Natural Gas Consumption Forecasting Model Based on Feature Optimization and Incremental Long Short-Term Memory

**DOI:** 10.3390/s25103079

**Published:** 2025-05-13

**Authors:** Huilong Wang, Xianjun Gao, Ying Zhang, Yuanwei Yang

**Affiliations:** 1Wuhan Jiyi Network Technology Co., Ltd., Wuhan 430205, China; wanghuilong@geetest.com (H.W.); zhangying@geetest.com (Y.Z.); 2School of Geosciences, Yangtze University, Wuhan 430100, China; 516042@yangtzeu.edu.cn

**Keywords:** natural gas, forecasting, autoregressive, neural networks, demand and supply, adaptive forecasting models, LSTM

## Abstract

Natural gas, as a vital component of the global energy structure, is widely utilized as an important strategic resource and essential commodity in various fields, including military applications, urban power generation and heating, and manufacturing. Therefore, accurately assessing energy consumption to ensure a reliable supply for both military and civilian use has become crucial. Traditional methods have attempted to leverage long-range features guided by prior knowledge (such as seasonal data, weather, and holiday data). However, they often fail to analyze the reasonable correlations among these features. This paper proposes a natural gas consumption forecasting model based on feature optimization and incremental LSTM. The proposed method enhances the robustness and generalization capability of the model at the data level by combining Gaussian Mixture Models to handle missing and anomalous data through modeling and sampling. Subsequently, a weakly supervised cascade network for feature selection is designed to enable the model to adaptively select features based on prior knowledge. Finally, an incremental learning-based regression difference loss is introduced to promote the model’s understanding of the coupled relationships within the data distribution. The proposed method demonstrates exceptional performance in daily urban gas load forecasting for Wuhan over the period from 2011 to 2024. Specifically, it achieves notably low average prediction errors of 0.0556 and 0.0392 on the top 10 heating and non-heating days, respectively. These results highlight the model’s strong generalization capability and its potential for reliable deployment across diverse gas consumption forecasting tasks within real-world deep learning applications.

## 1. Introduction

In recent years, with the rapid development of industrial technology, the demand for natural gas among residents has been steadily increasing. However, due to the influence of weather changes, market demand, and other factors, accurately predicting the daily residential natural gas load has become a significant challenge for effectively assessing energy consumption [1,2,3,4,5,6]. Previously, some researchers used traditional statistical models [7,8,9], such as the Autoregressive Integrated Moving Average (ARIMA) [10,11] model and regression models, to perform nonlinear modeling of natural gas data. However, these simple nonlinear relationships were insufficient to model the incremental, periodic, and coupled features within data sequences.

As a primary component of the global energy mix, natural gas has accounted for nearly 25% of worldwide energy consumption in recent years [12]. This trend reflects the rapid advancement of industrial technologies and the ongoing transition toward cleaner, more affordable, abundant, and cost-effective energy sources, which are gradually replacing conventional fossil fuels [13]. With the increasing demand for natural gas for both industrial operations and residential usage, ensuring a reliable supply hinges critically on accurate consumption forecasting [14]. However, current gas demand forecasting methods face three major challenges [14,15,16]: (1) Temporal sensitivity: Daily gas consumption exhibits substantial intra-day fluctuations due to usage peaks. Furthermore, seasonal variations cause extreme distributions in summer and winter. Additionally, long-term forecasting is often coupled with multi-energy systems, complicating the prediction due to cross-energy interactions. (2) Spatial heterogeneity: Gas consumption patterns vary significantly across regions (e.g., industrial zones vs. residential areas) and user types (e.g., industrial boilers, residential distributed heating, and CNG refueling stations), leading to complex spatial dynamics. (3) Multi-factor coupling effects [17,18]: Gas demand is dynamically influenced by multiple exogenous factors, including extreme weather events, public holidays, and macroeconomic cycles. Accurately forecasting natural gas demand is, therefore, essential, not only to improve operational efficiency, reduce costs, and conserve energy but also to support reliable supply and avoid overconsumption. Motivated by these challenges, residential daily natural gas consumption forecasting has garnered substantial attention from both researchers and industry practitioners. Commonly adopted evaluation metrics include the Mean Absolute Error (MAE) and Mean Absolute Percentage Error (MAPE) [19,20] for regression accuracy. Moreover, to assess the model’s temporal robustness, metrics such as advance warning accuracy (i.e., the ability to forecast threshold fluctuations N hours in advance) and response error under extreme conditions are also employed.

Subsequently, machine learning methods such as Support Vector Machines (SVMs) [21], K-Nearest Neighbors (KNN) [22], decision trees [23], and ensemble learning [24] were introduced. These methods successfully fit coupled data through high-order nonlinear modeling but still faced challenges when handling time-series information. With the development of deep learning across various domains, promising accuracy levels have been achieved. In the field of natural gas forecasting, the ARIMA model, as a classic regression model, first uses historical data for prior predictions. It then applies differencing to smooth the data, effectively addressing the relationships among strongly coupled data. Finally, cumulative error corrections are used to refine the model, overcoming the inability of direct regression models to capture long-range dependencies in sequences. However, frequent smoothing operations during the model training stage significantly reduce the interaction among features. Subsequently, Baldacci et al. [25] sampled historical data to construct sample models, using KNN and local regression algorithms for prior regression modeling, achieving notable prediction accuracy in both rural and urban settings. Gorucu et al. [26] predicted natural gas consumption using conditional control variables (such as temperature, natural gas prices, customer count, and exchange rates).

Machine learning, with its excellent data-fitting capabilities, has been widely applied to various regression forecasting tasks. Common methods include Support Vector Regression (SVR), Artificial Neural Networks (A NNs) [27], and Recurrent Neural Networks (RNNs [28]). Among these, RNNs, with their chain-like structure and memory loops, efficiently learn feature distributions from time-series data. Long Short-Term Memory networks (LSTM [28,29,30]), a special form of RNNs, have been extensively used in energy consumption forecasting tasks due to their powerful context feature-capturing ability. For example, localized LSTM models can be trained and predicted based on different feature subsets (e.g., season, temperature, and user type), improving model adaptability and accuracy. Labi et al. [30] developed a composite LSTM recurrent model based on consumption profile recognition for day-ahead natural gas consumption forecasting. Moreover, some research has combined LSTM with Convolutional Neural Networks (CNNs) to leverage the CNN’s spatial invariance for local feature extraction [31,32,33], complementing LSTM’s capability to capture long-range features. LSTNet [34] further introduced a skip-RNN to handle long-term temporal features and added an autoregressive component to enhance the constraint between input and output predictions. Subsequently, Temporal Convolutional Networks (TCNs) [35] optimized both an RNN and CNN, achieving a balance between training time and inference speed. However, the above methods largely overlooked the strong temporal coupling characteristics of natural gas forecasting. The natural gas daily load exhibits stochastic variations influenced by time factors such as seasons and holidays.

In summary, existing methods primarily focus on short-term data for supervised prediction without fully exploring the impact of external factors such as seasonality, holidays, and weather. Notably, these external factors exhibit significant sensitivity to regression predictions. Current approaches are limited to performing time convolution operations within a local feature range, which restricts their ability to analyze local features over short periods. They often overlook the long-term effects of external factors like holidays and weather, making them unsuitable for predicting irregular and sensitive data and leading to delays in prediction responses. A recent study, DCSDNet [12], takes into account the multiple seasonalities and irregularities of natural gas consumption. It employs a dual-convolution seasonal analysis network, which first uses MSTL (Multi-Seasonal Time-series Decomposition) to analyze the effects of various seasonalities on regression predictions. Subsequently, it integrates local and global convolutional networks to extract short-term and long-term features from data sequences for mathematical modeling. Finally, it incorporates an autoregressive model to predict historical consumption, further enhancing the coupling between historical and future predictive features. However, as the prediction range increases, the accuracy of the predictions diminishes under the influence of unexpected events within the prediction window, such as holidays and weather changes [36]. Later, Na et al. [37] proposed a Wavelet LSTM model based on wavelet transformation and Long Short-Term Memory networks. This model preprocesses feature data using wavelet transformation to construct data associations. Validation on real natural gas datasets demonstrated that Wavelet LSTM outperformed traditional LSTM in understanding the coupling relationships among sequence features and improved prediction accuracy. Shu et al. [38] analyzed daily and quarterly natural gas load characteristics in Chengdu and proposed using an iterative weakly supervised learner (XGBoost [39]). By employing a custom loss function during data training, their model was guided to adapt to multitask learning scenarios effectively.

Despite the methods mentioned above attempting to fit features through feature-guided approaches, such fitting often lacks correlations among features. Specifically, the daily natural gas load is influenced not only by historical data but also by external factors, such as seasons and weather. Designing an adaptive feature selection mechanism remains a key challenge in addressing daily natural gas load forecasting. To tackle this issue, we propose a natural gas forecasting model based on feature optimization and incremental LSTM. Specifically, we first utilize a Gaussian Mixture Model to handle missing and anomalous data through modeling and sampling, thereby enhancing the robustness and generalization capability of the model. Next, we design a weakly supervised cascade feature selection model to adaptively perform prior feature selection for daily natural gas loads. Subsequently, LSTM is employed to incrementally predict the prior-selected features. Incremental learning adjusts network gradients adaptively using weather, holidays, and total natural gas consumption as scaling factors. This encourages the model to explore potential coupling relationships within the data during the training phase, further enhancing its generalization capability.

In summary, the main contributions of this paper are as follows:(1)Proposed a method for modeling missing/anomalous data sampling based on Gaussian mixture distributions, enhancing the robustness and generalization capability of the network;(2)Developed a weakly supervised cascade network feature selector, which improves the coupling between historical and current data, enabling the network to achieve more accurate predictions when dealing with sensitive data;(3)Introduced an incremental prediction method to address the prediction lag caused by anomalous data, fully leveraging the model’s feature-awareness capabilities;(4)Lastly, designed a regression difference loss in conjunction with the mean squared error loss to enhance the feature fusion of long-range data associations, improving the accuracy of natural gas consumption forecasting.

The remainder of this paper is organized as follows: Section 2 presents the architectural design of the proposed method, with each subsection detailing the design motivation and principles. Section 3 describes the dataset sources and experimental setup, followed by a quantitative and qualitative analysis of the experimental results. Subsequently, ablation studies are conducted to validate the superiority of our approach. Finally, Section 4 provides a detailed discussion and comprehensive conclusions of our method. A summary of all abbreviations and symbols used in this work is provided in Table 1 for clarity and reproducibility.

## 2. Materials and Methods

In this subsection, we provide a detailed description of our method, along with an explanation of the design motivations behind it. As shown in Figure 1, we describe the overall network framework. Subsequently, we elaborate on the relevant gas forecast techniques employed in our method. Finally, we give a brief overview of the loss functions used in our approach.

### 2.1. Data Process

Previous models, such as LSTM and attention mechanism-based LSTM, have attempted to deepen the network layers to achieve more advanced feature selection. However, deeper networks often result in the loss of shallow feature representations, making it challenging to mathematically model the nonlinear relationships of long-sequence features in natural gas consumption forecasting effectively. To address this issue, we first preprocess the anomalous and missing data in the natural gas dataset. The specific processing method is shown in Equation (1).(1)$$X^{\left(l\right)}=Adaptive(\frac{X^{\left(l\right)}-{X_{min}}^{\left(l\right)}}{{X_{max}}^{\left(l\right)}-{X_{min}}^{\left(l\right)}})$$
where $X^{\left(l\right)}$ represents the length of the feature sequence, and adaptive (Equation (4)) is a function used for adaptively removing anomalous and missing data, while $X_{max}$ and $X_{min}$ represent the maximum and minimum values in the dataset, respectively.

For deep neural networks (DNNs) [40], preventing overfitting and enhancing the model’s robustness are critical steps that must be prioritized in natural gas consumption forecasting. Previous models, such as KS-LSTM [41,42], attempted to predict missing values to generate pseudo-predictions, which were then used to control the gradient of the model discriminatively, thereby improving the model’s robustness and generalization capability. In this paper, we construct a Gaussian Mixture Model (GMM) to establish the relationships between anomalous and missing data. A GMM is a typical probabilistic model used to represent mixed distributions. A one-dimensional GMM consists of *K* components, where each component represents one of the *K* mixed distributions. The formula for the GMM distribution is shown below. This approach allows us to utilize the constructed data for model training, thereby enhancing the model’s generalization capability.(2)$$g(x,u,\sigma )=\frac{1}{\sqrt{2\pi \sigma^{2}}}e^{-\frac{{\left(x-u\right)}^{2}}{2\sigma^{2}}}$$
Here, *x* represents the input distribution, while u and σ denote the mean and variance of the distribution, respectively. Therefore, the formula for the Gaussian Mixture Model is shown as Equation (3).(3)$$G(x,u,\sigma )=\sum_{i=0}^{K-1}\left(x_{i},u_{i},\sigma_{i}\right)$$
where *i* represents each Gaussian distribution.

We apply a Gaussian Mixture Model (GMM) to the anomalous and missing data belonging to the same category. Subsequently, feature sampling is performed following a uniform distribution. The uniform distribution further eliminates central bias errors, enabling the model’s distribution to better align with real-world scenarios.(4)Adaptive(x(l))(a,m)=Random(∑i=0K−1G(x(a,m),(i)x(a,m).(i)mean,x(a,m).(i)var),l×r)

Specifically, uniform sampling is performed within the range [x(a,m).(i)mean−x(a,m).(i)var,x(a,m).(i)mean+x(a,m).(i)var], where the mean and standard deviation correspond to the center of the Gaussian distribution for features of the same category. When the number of features for the same category is less than l×r, all available features are used for training. Conversely, when the number of features exceeds l×r, features are randomly sampled within the specified range based on the aforementioned sampling principle.

Here, *l* represents the feature sequence of the same category, *K* denotes the feature categories, and *r* is the feature selection factor, which is set to 0.5 in this study. This indicates that abnormal data features (Abnormal: a) and missing data features (Miss: m) are uniformly selected alongside general data features to perform soft supervised learning on the model. These operations further enhance the robustness of the model.

### 2.2. Weakly Supervised Cascade Network Feature Selection

Previous methods have focused on constructing deep learning models for single-feature selection, often overlooking the correlations between historical and current data [43,44]. These correlations are particularly evident in how time, seasonal holidays, and specific events or activity days influence daily natural gas load variations. Therefore, by observing historical data, we can make a rough prediction of the current day’s natural gas load. Motivated by this, we leverage historical data, considered as weakly labeled data, to select features for the regression prediction model and establish a nonlinear relationship. As shown in Figure 2, we implement a multi-model voting mechanism to perform feature selection on redundant multi-input features.

(1)First, for the input samples, we construct the feature mapping relationship by minimizing the Gini index criterion. The Gini index is shown in Equation (5):
(5)$$gini(f^{l})=1-\sum_{k=0}^{l-1}(\frac{f^{\left(j\right)}}{\sum_{j=0}^{j=i-1}f^{\left(j\right)}}{)}^{2}$$
where *l* represents the length of the feature, and *k* and *j* represent feature sequences.

(2)Subsequently, we use a multi-cascade network to evaluate feature responses in the dataset. Initially, the model analyzes various factors such as time, holidays, special events, and school schedules to establish feature mappings that influence the daily natural gas load. To address uncertainties in feature representation, we incorporate an uncertainty gradient into the mean squared error loss, enabling fine-tuning of the model’s parameters. This approach helps the model adapt effectively to datasets with similar characteristics, enhancing its feature selection capability. By iteratively refining this process, the model constructs highly responsive relationships for daily natural gas load forecasting. Equation (6) provides a detailed description of this process:
(6)$$\begin{array}{l} loss(x,p,t)=loss_{supervised}\times loss_{unsupervised}\\ =\frac{1}{N}{\sum_{b=0}^{i-1}\left(p_{i}-t_{i}\right)}^{2}+\sum_{b=0}f_{\Delta }\times \frac{\partial model(x_{b};\theta )}{\partial \theta } \end{array}$$
where *x* represents the input data, $loss_{supervised}$ is the mean squared error loss, $loss_{unsupervised}$ is the gain unsupervised loss, *N* is the total number of samples, and *b* represents a single training sample.

**Figure 2 sensors-25-03079-f002:**
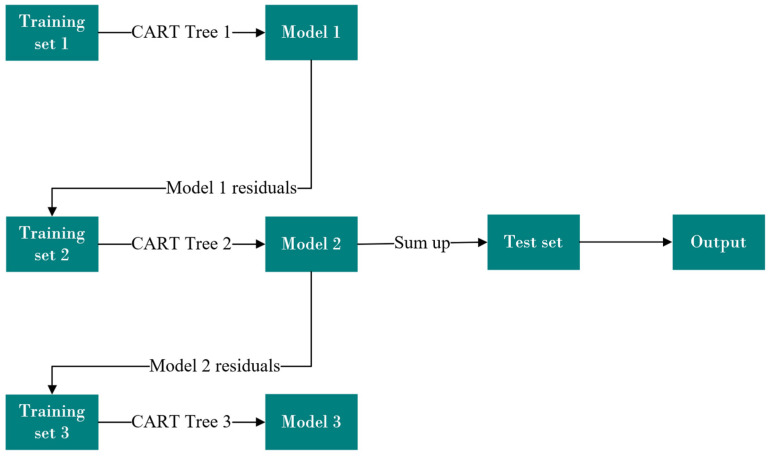
Weakly supervised cascading network feature selection framework.

Using the above method, the XGBOOST model was used to filter all features originally included, and finally, the type main features, as shown in Table 2, were selected.

### 2.3. LSTM Network Architecture

After performing high-responsive feature selection, constructing an accurate regression model becomes critical. Traditional RNNs are unable to capture long-distance sequential feature information effectively. LSTM (see Figure 3), through its gating mechanisms (input gate, forget gate, and output gate), efficiently facilitates the interaction between short-range adjacent feature information and long-range global feature information.

(1)Input gate: For the input data xt∈ℝL, in our time-series setting, the input sequence is denoted as {x1, x2, …, xt}, where each xt represents the feature vector at a time step, where it is first passed through an activation function for feature regularization, resulting in y^t, as shown in the following formula:
(7)y^t=Sigmoid(wi,t⋅cat(xt,ht−1)+bi)t=11+e−(wi,⋅cat(xt,ht−1)+bi)
where *t* represents the time-series state, $w_{i}^{t}$ denotes the weight value at a specific time step, and $b_{i}^{t}$ represents the bias value at a specific time step.

(2)Forget gate: By performing cascade network feature selection under weak supervision, the forget gate can better respond to high-weight features while suppressing the interference of disruptive features on the network.
(8)$$f_{t}=Sigmoid(w_{f}\cdot cat(x_{t},h_{t-1})+b_{f})$$
Here, $w_{f}$ represents the output weight of the network node from the previous stage, and $b_{f}$ is the learnable bias.

(3)Output gate: Features are interacted through short-distance skip connections, effectively preventing gradient vanishing.
(9)$$\begin{array}{l} O_{t}=Sigmoid({w_{i,}}^{t}\cdot cat(x_{t},h_{t-1})+b_{o})\\ h_{t}=O_{t}\times tanh(C_{t}) \end{array}$$
Here, $b_{o}$ is initialized to 1, and $C_{t}$ represents the learnable parameter.

### 2.4. Incremental Learning for Regression Prediction

Due to the influence of external factors such as holidays and weather, traditional LSTM models struggle to handle data distribution anomalies caused by sensitive data, leading to prediction lag and significant declines in regression accuracy. In this paper, we propose Incremental Learning for Regression Prediction, which enhances the coupling between sequential data by introducing additional regression variables to address the issue of declining prediction accuracy as the prediction time range increases. Specifically, we add fully connected layers to the LSTM model to improve its incremental regression prediction capability. This process is defined as follows:

Traditional LSTM-based models often suffer from prediction lag and performance degradation when confronted with significant shifts in external environmental factors, such as holidays or abrupt weather changes. To address this limitation, we propose an Incremental Learning for Regression Prediction strategy, which substantially enhances the model’s adaptability to distributional anomalies and improves forecasting accuracy under non-stationary conditions. Specifically, in the first stage, the model takes an input data sequence x and generates predictions ${\hat{y}}_{t}$ across multiple time horizons. The differences between adjacent time steps are then computed, as shown in Equation (10), to obtain the true incremental signals, with the temporal span in this study set to t=2. In the second stage, a nonlinear fully connected layer is designed to perform incremental regression prediction based on the generated outputs ${\hat{y}}_{t}$, enabling the model to learn fine-grained temporal changes in the target series. Joint optimization of the base forecasting task and the incremental prediction objective enables the model to better capture the intrinsic dependencies within sequential data. This coupled learning framework enhances robustness and effectively mitigates the decline in prediction accuracy caused by missing or anomalous inputs. To further enhance the generalization capability of the proposed Incremental Learning for Regression Prediction strategy, we incorporate real-time monitoring of the incremental prediction residuals (as defined in Equation (12)). When the residual consistency metric ζ remains below a predefined threshold (0.5) for consecutive steps, a localized fine-tuning process is activated. In this phase, only the fully connected layers responsible for incremental regression are updated, while the LSTM backbone remains frozen to preserve the model’s stability. Moreover, during training, Gaussian noise—with a standard deviation set to 10% of the current batch’s standard deviation—is added to simulate disruptions caused by external factors such as extreme weather or holidays. This augmentation improves the robustness of the model in handling unexpected or rare events.(10)$$\nabla_{gt}(x)=(LSTM{\left(x\right)}_{t}-LSTM{\left(x\right)}_{t-1})$$(11)$$\nabla_{p}=Linear_{1\times }(LSTM(Linear_{1\times }(1,dim)),1)$$(12)$$\zeta =|\nabla_{gt}-\nabla_{p}|$$

Herein, $\nabla_{p}$ denotes the model-generated incremental prediction, and $\nabla_{gt}$ represents the corresponding ground-truth increment, using the loss function from Section 2.5. $Linear_{n\times }$ denotes the fully connected layer, where n denotes the number of fully connected layers, which is set to 1 in this paper. LSTM refers to the LSTM model from Section 2.3, and dim represents the number of neurons in the hidden layer.

### 2.5. Incremental Loss Function

Traditional regression models typically use mean squared error (MSE) loss to construct regression relationships for supervision. However, in the field of natural gas forecasting, MSE loss is insufficient for effectively supervising the correlations between data points. In this paper, we design a regression difference loss $L_{r}$ in conjunction with MSE loss to fully leverage the differences between predicted values for regression supervision, as defined below:(13)$$L_{r}(\nabla^{\left(l\right)};\theta )=\frac{1}{l}\sum_{i=0}^{l-1}{\left(\nabla_{p}-\nabla_{gt}\right)}^{2}$$
where *l* represents the length of the feature sequence, and $\nabla_{p}$ and $\nabla_{gt}$ are defined as shown in Equations (10) and (11).

Finally, the overall loss L of our model is defined as follows:(14)$$L=\frac{1}{N}{\sum_{i=0}^{l-1}\left(y-{\hat{y}}_{t}\right)}^{2}+L_{r}$$ Here, *y* represents the label value.

Efficient data filtering provides a solid data foundation for enhancing the robustness of the model. Subsequently, the weakly supervised cascade feature selection identifies key features relevant to natural gas forecasting. Finally, the long-range data correlation modeling loss (regression difference loss) promotes the accuracy of natural gas forecasting.

## 3. Experiment and Result Analysis

### 3.1. Data Description

This study leverages multi-source data collected from the SCADA system of Wuhan’s high-pressure natural gas pipeline network, hourly meteorological observations from the Wuhan Meteorological Bureau, and officially published national holiday calendars by the General Office of the State Council. The dataset spans from 30 December 2010 to 5 January 2025, covering three complete annual consumption cycles and supporting long-term pattern learning. Daily gas consumption is derived from real-time measurements recorded by Rosemount 3051S differential pressure flowmeters deployed across 13 administrative districts in Wuhan, including 7 industrial zones and 6 residential areas. The sensors report instantaneous flow every 5 min, which is aggregated into daily totals (unit: 10,000 m^3^/day). To ensure data integrity, periods with abnormal pressure values (<0.2 MPa or >4.0 MPa) are automatically discarded, and 0.42% of missing entries are imputed via temporal linear interpolation, preserving the continuity of time-series inputs for model training.

Our analysis of gas consumption patterns reveals that industrial usage exhibits a distinct bimodal distribution, with peaks occurring between 08:00 and 10:00 and 18:00 and 20:00. The peak-to-valley ratio reaches 1:2.3, and weekday consumption is 1.8 times higher than that of weekends. Residential gas consumption is significantly affected by seasonal factors: the average daily usage in the winter is 3.1 times that in the summer, and daily volatility during the heating season increases by 47% compared to non-heating periods.

Meteorological data integrate hourly observations (temperature, humidity, and wind speed) with 72 h temperature forecasts. Through feature engineering, three categories of variables are constructed: (1) cumulative cold effect: when the daily average temperature is below 5 °C, gas consumption increases by 7.2% for each consecutive day; (2) temperature–humidity interaction: when the temperature is below 10 °C and humidity below 60%, each 1% drop in humidity results in a 1.8% increase in consumption; and (3) temperature forecast error compensation: when ∣ΔT∣ > 3 °C, a dynamic compensation module is triggered to correct the deviation.

Holiday data are encoded based on the national holiday calendar issued by the State Council, segmented into three phases: (1) pre-holiday stockpiling: during the 7 days before the Spring Festival, daily consumption increases by 4.5%; (2) holiday core period: industrial and commercial consumption decreases by 22%; and (3) post-holiday recovery: in the 3 days after the holiday, consumption gradually rebounds.

The generalizability of the proposed model is constrained by three main factors:(1)Climate dependency: the model is developed based on a subtropical monsoon climate (Köppen Cfa) and requires recalibration of temperature thresholds when applied to tropical or high-latitude regions.(2)Differences in gas consumption structure: in Wuhan, industrial usage accounts for 42% of total gas consumption, significantly higher than in many European and North American cities (typically <30%), which may lead to an overestimation of industrial demand when transferring the model.(3)Pipeline network topology: Wuhan adopts a looped gas network with 28% redundancy, whereas cities like Paris use a radial topology with only 12% redundancy, leading to different pressure response dynamics under the same load variations.

In addition, factors such as the seven-day Spring Festival holiday in China (with an average residential consumption drop of 8%) versus the Western Christmas holiday (with a 25% drop), as well as the “coal-to-gas” policy transition during 2017–2020, should be carefully considered for cross-regional model transfer.

### 3.2. Implementation Details

In this study, temperature, humidity, wind speed, atmospheric pressure, precipitation, time, and holiday information from the WHU Larger-Scale Pipeline Natural Gas Dataset (WNGD) are utilized as input features for both the proposed method and the baseline methods. The experiments are conducted on a hardware environment equipped with an NVIDIA GTX 4060Ti GPU and software environment consisting of Windows 10, Python 3.10, and PyTorch 1.18 with CUDA 12.1. For all models, the batch size is fixed at 64, and the random seed is set to 1234 to ensure reproducibility. Training (see Table 3) is performed for 5000 epochs using the Adam optimizer, with an initial learning rate of 1 × 10^−3^ and a learning rate decay of 1 × 10^−2^. The training process is organized into four distinct stages: rapid descent, parameter optimization, smooth fitting, and fine smoothing. An early stopping strategy is applied to prevent overfitting and to identify optimal parameter solutions. Furthermore, the validation loss on the test set is continuously monitored to ensure model generalization and to further mitigate overfitting. Furthermore, we included Table A1, Table A2, Table A3 and Table A4 in the Appendix A to provide detailed descriptions of the model architectures for both the baseline methods and our proposed approach. These tables include the input and output feature dimensions, the internal components of each model, and the types of activation functions used.

### 3.3. Evaluation Metrics

To evaluate the effectiveness of the proposed method, we used the Mean Absolute Percentage Error (MAPE) to assess the model’s accuracy. The calculation formula for the evaluation metric is as follows:(15)$$MAPE=\frac{1}{N}\sum_{i=0}^{N-1}\left|\frac{y_{i}-{\hat{y}}_{i}}{y_{i}}\right|\times 100$$
where *N* represents the number of samples, y is the true label, and $\hat{y}$ is the predicted value from the model.

### 3.4. Experimental Analysis

*(1) Experimental analysis of comparative methods:* Firstly, to further analyze the prediction errors and their underlying causes, we conducted both quantitative and qualitative comparisons between our proposed method and several baselines, including the Backpropagation Neural Network (BP), Transformer, and Temporal Fusion Transformer (TFT) [45] models, under different conditions. The results demonstrate that our method consistently outperforms the others in terms of both prediction accuracy and stability.

As shown in Table 4, the error growth rate of our method is significantly slower than that of the comparison models as the forecast horizon extends, indicating better stability in mid- to long-term forecasting tasks. For instance, when forecasting the top 10 days, the Absolute Mean Percentage Error (MAPE) of our method on heating and non-heating days reaches 0.0556 and 0.0392, respectively.

The BP model suffers from larger errors primarily due to its reliance on traditional gradient descent-based optimization without effective integration of multi-factor features relevant to gas consumption. Although Transformer and TFT architectures are capable of capturing long-range dependencies, their performance drops significantly during highly volatile periods (e.g., holidays) due to the absence of an effective feature selection mechanism. In contrast, our method integrates a weakly supervised cascad network feature selection and an Incremental Learning for Regression Prediction strategy, which substantially enhances the model’s capacity to model long-term dependencies and adapt to spatial heterogeneity. Furthermore, the introduced incremental loss contributes to higher predictive accuracy under complex conditions.

Figure 4 illustrates the temporal trend of daily prediction errors across different methods. It is evident that our model achieves lower initial prediction errors after incorporating the feature selection module, suggesting that a well-designed feature coupling strategy can help the model focus on key variables early in training, thereby improving convergence efficiency. During heating days, baseline models suffer from rapidly increasing errors as the prediction step grows, especially under strong feature coupling scenarios, resulting in steep error gradients and degraded inference performance. On non-heating days with more stable consumption patterns, all models exhibit better overall performance, but ours still maintains a leading edge. To analyze the learning stability and convergence of the proposed model, we present the loss trajectories on the training and validation sets (see Figure 5).

Finally, we performed a sensitivity analysis on key factors affecting prediction accuracy. The results highlight that heating status (heating vs. non-heating days), feature coupling intensity, sequence volatility (e.g., holidays), and forecast horizon are the most influential variables. Our method effectively addresses these challenges through carefully designed feature selection and model architecture, thereby improving overall predictive performance.

*(2) Analysis of prediction failures in the WNGD*: This subsection analyzes SHAP visualizations based on real weather data collected in Wuhan from 1 January to 9 February 2025, in conjunction with model prediction errors for both the HDD and NHD. As illustrated in Figure 6, the prediction errors are primarily influenced by temperature, humidity, wind score, and precipitation. Specifically, for HDD prediction, the error tends to decrease with increasing atmospheric pressure, whereas for NHDs, the error increases with rising pressure and humidity. Furthermore, as shown in Table 4, the prediction errors from day 28 to day 35 exhibit higher stability compared to earlier periods, with average error values of 0.104 and 0.044 for HDDs and NHDs, respectively. These results suggest that features such as pressure, humidity, and time effectively support the regression performance of the proposed model. Notably, on day 28, the prediction errors for the HDD and NHD reached 0.115 and 0.048, which were 0.005 and 0.006 higher than those on day 27 when holiday-related features were not incorporated. Following the inclusion of holiday indicators, the errors gradually declined. This improvement can be attributed to two key mechanisms: (1) the weakly supervised cascad feature selection strategy that leverages auxiliary features for short-term regularization to mitigate erroneous predictions, and (2) the incremental loss optimization, which exploits historical prediction features to enhance long-term memory and decision-making capability. The synergy of these strategies enables the model to retain robustness and generalization performance even when encountering atypical or erroneous cases.

### 3.5. Ablation Study

*(1) Effectiveness of the loss function:* From Figure 7 and Figure 8, we can clearly observe that in short-term daily natural gas load forecasting, the MAPE gradient increases significantly more slowly with prediction time when using regression difference loss compared to not using it. This improvement is attributed to the regression difference loss working synergistically with the mean squared error loss to establish an efficient data correlation supervision mechanism. Additionally, an interesting phenomenon is observed: when using regression difference loss, the error values for different types of data (heating days/non-heating days) remain below 0.06 during the first 15 days (see Table 5) of prediction. In contrast, without regression difference loss, the error values consistently exceed 0.06. This further demonstrates that regression difference loss has a clear advantage in handling strongly coupled time-series data, enabling efficient supervised utilization of the data.

*(2) Effectiveness of feature selection:* As shown in Figure 9, we compare the MAPE trends before and after applying feature selection, highlighting the performance improvement brought by filtering out less informative features, with the change in the number of features, and the results of all features (All features) show a relatively obvious upward trend, especially when the number of features is large, where the results keep improving, while the results of HDDs and NHDs present different growth trends, respectively. The upward trend of HDDs is relatively more significant, while that of NHDs is more stable. Overall, the selected features have a more significant improvement effect on enhancing the prediction performance of HDDs.

*(3) Effectiveness of adding future features:* Observations from Figure 10 and Figure 11 reveal that after applying L1 features, as shown in Table 6, the improvement in model prediction accuracy is significant, particularly in heating day load forecasting. By introducing L1 features, the model better captures the trends in load variations. This is especially evident in the early stages of prediction, where errors decrease substantially, and fluctuations stabilize. In contrast, without feature selection, the error for heating days gradually increases with the number of prediction days, highlighting the model’s inadequacy in capturing trends during long-term forecasting. For non-heating days, the overall error remains low and stable, with minimal differences between the two cases. This indicates that the load variation is relatively small and less dependent on future features.

*(4) Prediction comparison of weather features with different durations:* From the Figure 12, it is evident that after incorporating weather features of varying durations, the average MAPE for heating days is significantly higher than that for non-heating days. Furthermore, as the duration of the included weather features increases, the error for both heating and non-heating days exhibits an upward trend, with the error growth being more pronounced for heating days. This indicates that weather features have a substantial impact on natural gas load forecasting for heating days, contributing to higher prediction complexity and uncertainty. Gas load on heating days is strongly influenced by weather factors such as temperature, humidity, and wind speed. As weather features evolve over time, they may introduce greater volatility, leading to an accumulation of errors. In contrast, natural gas load forecasting for non-heating days shows better stability and robustness, with lower errors that increase more gradually. The natural gas demand on non-heating days is relatively stable and less influenced by weather factors, resulting in lower prediction errors.

*(5) Ablation study for the sensitivity of the adaptive feature sampling function (Equation (4)) with respect to the value of r:* As shown in Table 7, the sensitivity analysis on the value of r reveals that when *r* < 0.5, the prediction error increases consistently with longer forecasting horizons for both HHDs and NHDs. Nevertheless, the overall trend indicates that increasing *r* generally leads to lower prediction errors. Notably, when *r* = 0.8 or *r* = 1.0, the prediction error increases compared to *r* = 0.5 and values below. The optimal performance is achieved when *r* = 0.5, yielding the lowest overall prediction error. These findings suggest that in adaptive feature sampling, too few features can reduce the model’s ability to capture feature interactions at the sample level, while excessive features introduce noise that hinders the model’s fitting capacity. This further confirms that an appropriate sampling ratio enhances the model’s understanding of sample data, thereby improving its generalization and predictive performance.

## 4. Conclusions

To effectively address the challenge of low accuracy in daily natural gas load forecasting, this paper proposes a feature-optimized and incremental LSTM-based prediction frame, grounded on a natural gas consumption dataset from Wuhan, a representative city in Central China. First, we utilize a Gaussian Mixture Model to discretely model anomalous and missing data, selecting specific data to enhance the model’s generalization and feature-awareness capabilities. Subsequently, to improve the understanding of coupled features within natural gas data, we incorporate sensitivity-aware mechanisms to capture distributional shifts. This strategy proves effective for continuous learning and adaptation in cross-regional migration and long-term deployment scenarios. In addition, a weakly supervised cascade network for feature selection was utilized, improving the transferability and robustness across multi-scale spatiotemporal domains. Finally, to further promote long-range data correlations, we introduce a regression difference loss to efficiently supervise the fitting data and expanding its practical applicability over wider spatial ranges. Beyond natural gas forecasting, the proposed method demonstrates strong versatility in other energy-related applications. In renewable energy scenarios, our model integrated with multimodal data (e.g., wind speed, solar irradiance, and device states) effectively addresses the strong volatility of wind and solar power generation, offering disaster-resilient predictions. In the smart grid context, feature selection enables lightweight deployment, while incremental forecasting supports millisecond-level responses to electricity price signals for real-time optimization. Moreover, in cross-domain applications, the weakly supervised cascad network quantifies feature contributions, enabling the construction of collaborative control platforms for temporal energy forecasting across different sectors.

Despite its outstanding performance on the Wuhan dataset, the generalizability of the proposed natural gas forecasting model is subject to several regional constraints. Factors such as the geographical location, climate conditions, population density, energy structure, user types (residential, industrial, and commercial), local policies, and real-time gas prices all significantly influence gas consumption patterns. For example, in colder regions such as Russia, Canada, or Northern Europe, winter gas demand is highly volatile and strongly seasonal. In contrast, in tropical coastal areas, natural gas is mainly used for commercial cooling or industrial processes, with less seasonal fluctuations. Therefore, localized adjustments in feature engineering and coupling strategies are necessary to ensure adaptability. In addition, variations in energy supply systems, market mechanisms, and user behaviors across countries introduce further modeling challenges. For instance, China’s tiered pricing system may induce abrupt changes in user demand, affecting the stationarity and predictability of the time series. To address these issues and improve model generalization, future work will incorporate additional multi-factor features (e.g., public holidays, heating policies, diurnal temperature range, and urbanization ratios) and employ regional normalization and dynamic feature weighting strategies to enhance the model’s capacity to understand complex local characteristics.

In summary, the proposed feature-optimized and incremental LSTM-based method achieves promising results on the current dataset and shows potential for wider application. Future research will focus on cross-regional validation, cross-domain extension, and real-world deployment to promote the method’s trustworthy application in broader energy forecasting scenarios.

## Figures and Tables

**Figure 1 sensors-25-03079-f001:**
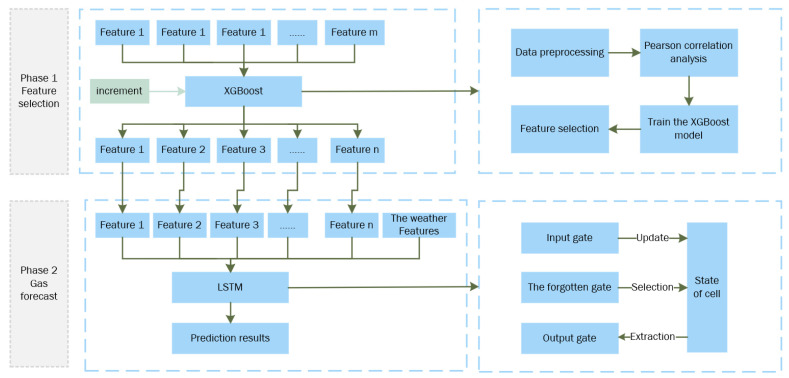
Overview of the proposed method.

**Figure 3 sensors-25-03079-f003:**
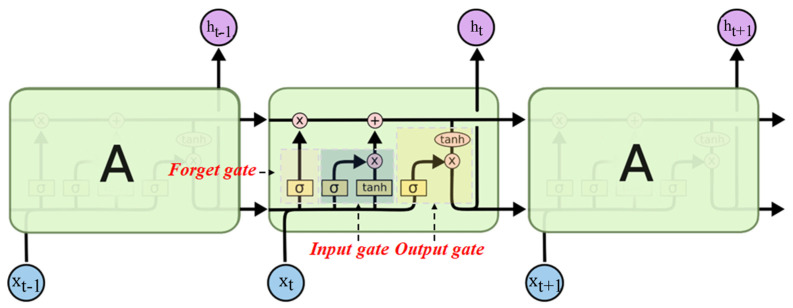
LSTM model structure.

**Figure 4 sensors-25-03079-f004:**
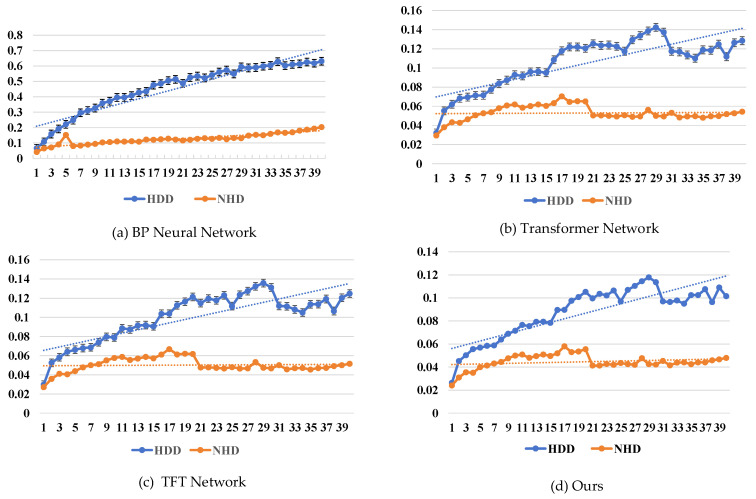
Comparison of predicted MAPEs across different forecast horizons between the proposed method and baseline approaches.

**Figure 5 sensors-25-03079-f005:**
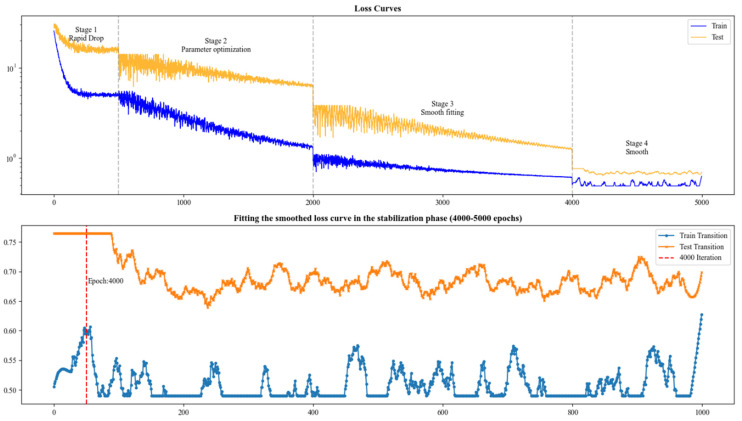
The loss curve during the training of our method is presented, where the entire process is divided into four distinct phases (as illustrated in the figure). The lower subfigure highlights the final smoothing and fitting stage.

**Figure 6 sensors-25-03079-f006:**
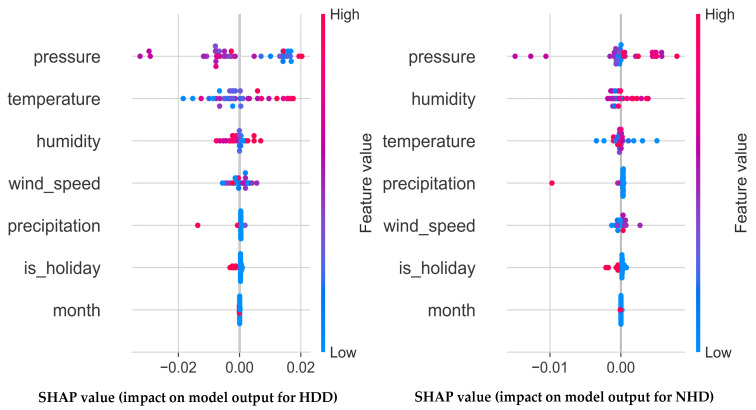
Visualization of the impact of input features on the prediction accuracy of HDDs and NHDs using SHAP.

**Figure 7 sensors-25-03079-f007:**
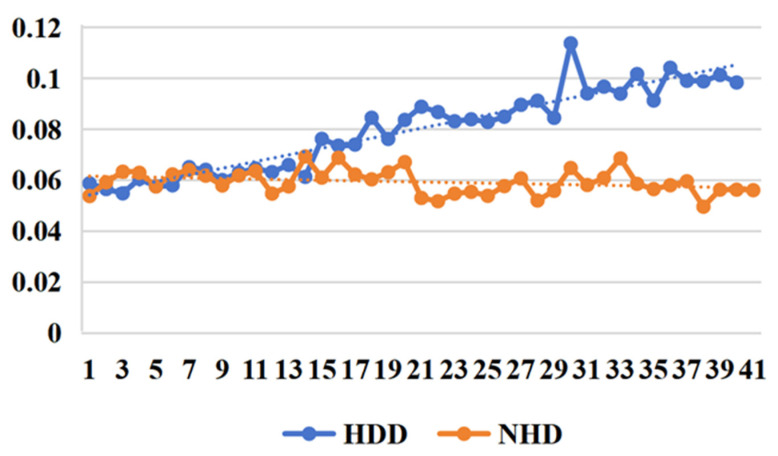
MAPE trend with predict day using no incremental loss.

**Figure 8 sensors-25-03079-f008:**
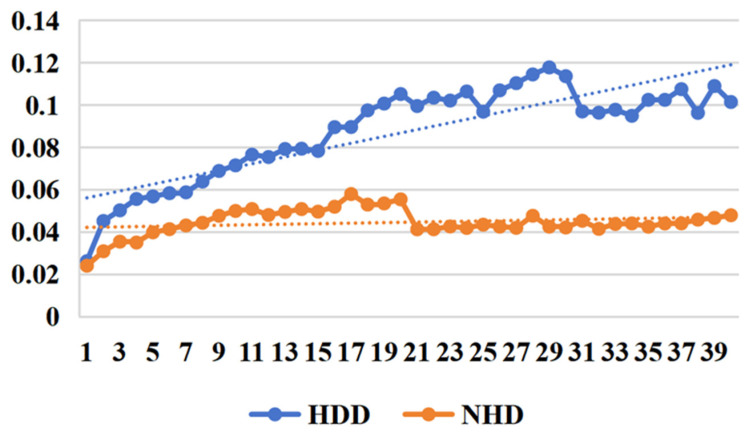
MAPE trend with predict day using our incremental loss.

**Figure 9 sensors-25-03079-f009:**
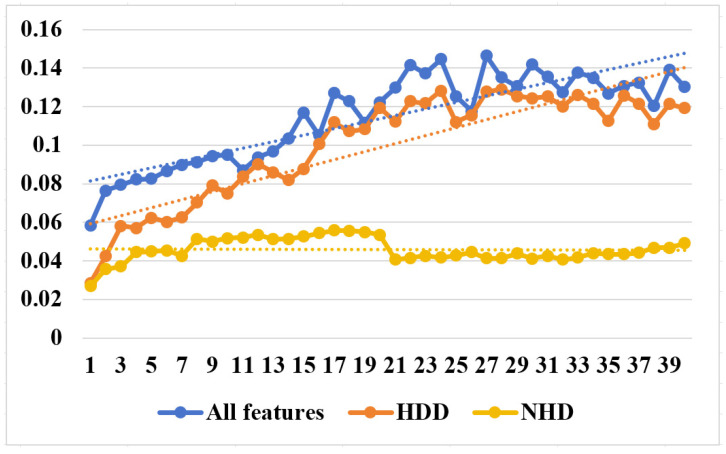
Comparison of MAPE trends before and after including all features and after filtering out the selected features.

**Figure 10 sensors-25-03079-f010:**
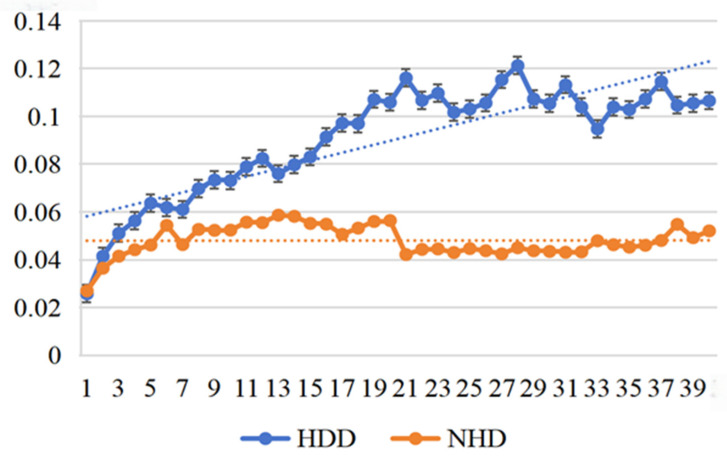
MAPE trend with predict day without adding L1 future date event features.

**Figure 11 sensors-25-03079-f011:**
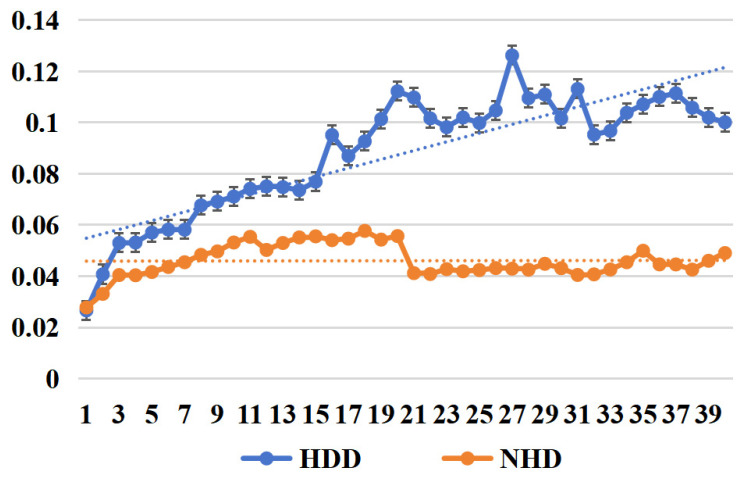
MAPE trend with predict day by adding L1 future date event features.

**Figure 12 sensors-25-03079-f012:**
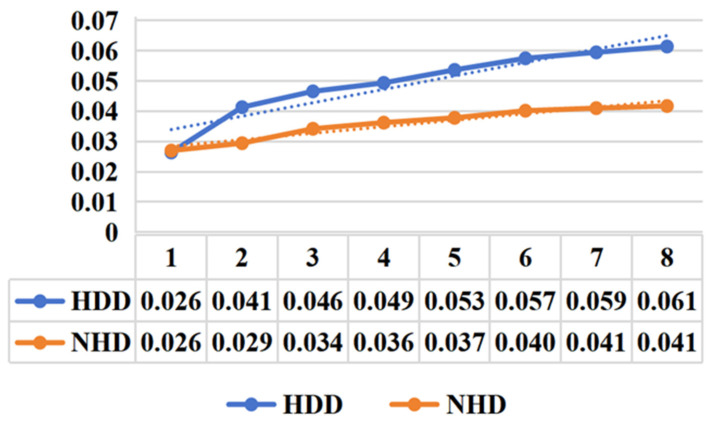
MAPE trend with predict day by different weather forecasting duration.

**Table 1 sensors-25-03079-t001:** List of abbreviations.

Category		Example
Acronyms	LSTM	Long Short-Term Memory
	ARIMA	Autoregressive Integrated Moving Average
	SVM	Support Vector Machine
	KNN	K-Nearest Neighbors
	ANN	Artificial Neural Network
	RNN	Recurrent Neural Network
	TCN	Temporal Convolutional Network
	MSTL	Multi-Seasonal Time-series Decomposition
	GMM	Gaussian Mixture Model
	MAPE	Mean Absolute Percentage Error
	BP	Backpropagation Neural Network
	HDD	Heating Degree Day
	NHD	Non-Heating Day
	TFT	Temporal Fusion Transformer
Abbreviations	et al.	And others
	Eq.	Equation
Symbols	*X*	Input feature
	*l*	Feature sequence length

**Table 2 sensors-25-03079-t002:** Different features at various levels (L1→L4, with each training dataset containing the features of the previous layer).

Feature Type	Features Related to the Daily Natural Gas Load					
Next date event (L1)	Time	Holiday	Special events	School		
Current date event (L2)	Time	Holiday	Special events	School		
Current monitoring features (L3)	Weather	Humidity	Wind speed	Precipitation	Heating status	
Previous monitoring features (L4)	Natural gas consumption	Air pressure	Heat index	Temperature variation	Humidity difference	Wind difference

**Table 3 sensors-25-03079-t003:** Experimental model parameter settings.

Model	Hyperparameters	Seed Initial	Train Strategy	Input Features	Dataset	Loss Functions
BP Neural Network	Batch size: 64 Optimizer: Adam Iteration: 5000 Init learning rate: 1 × 10^−3^ Decay rate: 1 × 10^−2^	1234	Early stopping	Temperature, humidity, wind speed, atmospheric pressure, precipitation, time, and holidays	WNGD	MSE loss
Transformer						
TFT						MSE loss + probabilistic loss
Ours						MSE loss + regression difference loss

**Table 4 sensors-25-03079-t004:** Quantitative MAPE results for comparative methods.

Method	Predict Date	1	2	3	4	5	6	7	8	9	10
BP Neural Network	HDD	0.066	0.108	0.158	0.191	0.221	0.248	0.297	0.311	0.327	0.358
	NHD	0.041	0.065	0.070	0.090	0.152	0.080	0.083	0.089	0.093	0.103
	Predict Date	11	12	13	14	15	16	17	18	19	20
	HDD	0.371	0.395	0.396	0.408	0.428	0.435	0.476	0.485	0.505	0.513
	NHD	0.105	0.110	0.109	0.111	0.108	0.122	0.121	0.124	0.127	0.122
	Predict Date	21	22	23	24	25	26	27	28	29	30
	HDD	0.487	0.525	0.534	0.522	0.541	0.561	0.573	0.549	0.594	0.587
	NHD	0.116	0.121	0.127	0.131	0.127	0.133	0.125	0.132	0.131	0.148
	Predict Date	31	32	33	34	35	36	37	38	39	40
	HDD	0.590	0.598	0.604	0.628	0.602	0.610	0.613	0.625	0.618	0.631
	NHD	0.153	0.150	0.159	0.169	0.166	0.169	0.180	0.186	0.194	0.203
Transformer	HDD	0.0321	0.0552	0.0621	0.0683	0.0695	0.0712	0.0715	0.0777	0.0837	0.0872
	NHD	0.0295	0.0381	0.0433	0.0428	0.0464	0.0506	0.0527	0.0539	0.0580	0.0608
	Predict Date	11	12	13	14	15	16	17	18	19	20
	HDD	0.0928	0.0917	0.0959	0.0963	0.0950	0.1086	0.1178	0.1219	0.1219	0.1205
	NHD	0.0619	0.0585	0.0602	0.0619	0.0604	0.0633	0.0704	0.0646	0.0653	0.0651
	Predict Date	21	22	23	24	25	26	27	28	29	30
	HDD	0.1253	0.1235	0.1239	0.1225	0.1174	0.1294	0.1337	0.1386	0.1423	0.1373
	NHD	0.0504	0.0505	0.0501	0.0493	0.0507	0.0491	0.0494	0.0562	0.0501	0.0494
	Predict Date	31	32	33	34	35	36	37	38	39	40
	HDD	0.1174	0.1169	0.1135	0.1101	0.1189	0.1184	0.1247	0.1118	0.1263	0.1285
	NHD	0.0532	0.04833	0.0494	0.0497	0.0481	0.0495	0.0498	0.0518	0.0527	0.0544
TFT	Predict Date	1	2	3	4	5	6	7	8	9	10
	HDD	0.0298	0.0526	0.0583	0.0641	0.0662	0.0679	0.0687	0.074	0.0797	0.0789
	NHD	0.0272	0.0359	0.0411	0.0406	0.0439	0.0479	0.0501	0.0513	0.0551	0.0577
	Predict Date	11	12	13	14	15	16	17	18	19	20
	HDD	0.0886	0.0874	0.0915	0.0919	0.0906	0.1037	0.1038	0.1126	0.1164	0.1213
	NHD	0.0588	0.0555	0.0571	0.0588	0.0573	0.0612	0.0669	0.0613	0.0621	0.0618
	Predict Date	21	22	23	24	25	26	27	28	29	30
	HDD	0.1149	0.1196	0.1178	0.1231	0.1117	0.1236	0.1275	0.1324	0.1358	0.1312
	NHD	0.0477	0.0478	0.0474	0.0467	0.0481	0.0466	0.0468	0.0534	0.0475	0.0468
	Predict Date	31	32	33	34	35	36	37	38	39	40
	HDD	0.112	0.1115	0.1083	0.1051	0.1136	0.1138	0.1192	0.1066	0.1204	0.1249
	NHD	0.0501	0.0458	0.0469	0.0472	0.0456	0.0471	0.0473	0.0492	0.0501	0.0516
Our network	Predict Date	1	2	3	4	5	6	7	8	9	10
	HDD	0.026	0.045	0.050	0.056	0.057	0.058	0.059	0.064	0.069	0.072
	NHD	0.024	0.031	0.035	0.035	0.040	0.041	0.043	0.044	0.048	0.050
	Predict Date	11	12	13	14	15	16	17	18	19	20
	HDD	0.077	0.075	0.079	0.079	0.078	0.090	0.090	0.098	0.101	0.105
	NHD	0.051	0.048	0.050	0.051	0.050	0.052	0.058	0.053	0.054	0.055
	Predict Date	21	22	23	24	25	26	27	28	29	30
	HDD	0.100	0.104	0.102	0.106	0.097	0.107	0.110	0.115	0.118	0.114
	NHD	0.041	0.041	0.043	0.042	0.044	0.043	0.042	0.048	0.043	0.042
	Predict Date	31	32	33	34	35	36	37	38	39	40
	HDD	0.097	0.096	0.098	0.095	0.102	0.102	0.108	0.096	0.109	0.101
	NHD	0.045	0.042	0.044	0.044	0.043	0.044	0.044	0.046	0.047	0.048

**Table 5 sensors-25-03079-t005:** Quantitative results by different loss functions.

Method	Predict Date	1	2	3	4	5	6	7	8	9	10
No incremental loss	HDD	0.059	0.057	0.055	0.061	0.058	0.058	0.065	0.064	0.060	0.063
	NHD	0.054	0.059	0.063	0.063	0.058	0.062	0.064	0.062	0.058	0.062
	Predict Date	11	12	13	14	15	16	17	18	19	20
	HDD	0.065	0.063	0.066	0.061	0.076	0.074	0.074	0.085	0.076	0.084
	NHD	0.064	0.055	0.058	0.069	0.061	0.069	0.062	0.060	0.063	0.067
	Predict Date	21	22	23	24	25	26	27	28	29	30
	HDD	0.089	0.087	0.083	0.084	0.083	0.085	0.090	0.091	0.085	0.114
	NHD	0.053	0.052	0.055	0.055	0.054	0.058	0.061	0.052	0.056	0.065
	Predict Date	31	32	33	34	35	36	37	38	39	40
	HDD	0.094	0.097	0.094	0.102	0.091	0.104	0.099	0.099	0.101	0.098
	NHD	0.058	0.061	0.069	0.059	0.057	0.058	0.060	0.050	0.056	0.056
Our incremental loss	Predict Date	1	2	3	4	5	6	7	8	9	10
	HDD	0.026	0.045	0.050	0.056	0.057	0.058	0.059	0.064	0.069	0.072
	NHD	0.024	0.031	0.035	0.035	0.040	0.041	0.043	0.044	0.048	0.050
	Predict Date	11	12	13	14	15	16	17	18	19	20
	HDD	0.077	0.075	0.079	0.079	0.078	0.090	0.090	0.098	0.101	0.105
	NHD	0.051	0.048	0.050	0.051	0.050	0.052	0.058	0.053	0.054	0.055
	Predict Date	21	22	23	24	25	26	27	28	29	30
	HDD	0.100	0.104	0.102	0.106	0.097	0.107	0.110	0.115	0.118	0.114
	NHD	0.041	0.041	0.043	0.042	0.044	0.043	0.042	0.048	0.043	0.042
	Predict Date	31	32	33	34	35	36	37	38	39	40
	HDD	0.097	0.096	0.098	0.095	0.102	0.102	0.108	0.096	0.109	0.101
	NHD	0.045	0.042	0.044	0.044	0.043	0.044	0.044	0.046	0.047	0.048

**Table 6 sensors-25-03079-t006:** Quantitative MAPE comparison results by adding L1 features.

Method	Predict Date	1	2	3	4	5	6	7	8	9	10
No feature selection	HDD	0.026	0.041	0.051	0.056	0.064	0.062	0.061	0.070	0.073	0.073
	NHD	0.027	0.036	0.042	0.044	0.046	0.054	0.046	0.053	0.052	0.052
	Predict Date	11	12	13	14	15	16	17	18	19	20
	HDD	0.079	0.082	0.076	0.080	0.083	0.091	0.097	0.097	0.107	0.106
	NHD	0.056	0.056	0.059	0.058	0.055	0.055	0.051	0.053	0.056	0.056
	Predict Date	21	22	23	24	25	26	27	28	29	30
	HDD	0.116	0.107	0.110	0.102	0.103	0.106	0.115	0.121	0.107	0.105
	NHD	0.042	0.044	0.045	0.043	0.045	0.044	0.042	0.045	0.044	0.044
	Predict Date	31	32	33	34	35	36	37	38	39	40
	HDD	0.113	0.104	0.095	0.104	0.103	0.107	0.115	0.105	0.106	0.106
	NHD	0.043	0.043	0.048	0.046	0.045	0.046	0.048	0.055	0.049	0.052
Our feature selection	Predict Date	1	2	3	4	5	6	7	8	9	10
	HDD	0.027	0.041	0.053	0.053	0.057	0.058	0.058	0.068	0.069	0.071
	NHD	0.028	0.033	0.041	0.040	0.042	0.044	0.045	0.048	0.0500	0.053
	Predict Date	11	12	13	14	15	16	17	18	19	20
	HDD	0.074	0.075	0.075	0.074	0.077	0.095	0.087	0.093	0.101	0.112
	NHD	0.055	0.05	0.053	0.055	0.056	0.054	0.055	0.058	0.054	0.056
	Predict Date	21	22	23	24	25	26	27	28	29	30
	HDD	0.110	0.102	0.098	0.102	0.100	0.105	0.126	0.11	0.111	0.102
	NHD	0.041	0.041	0.043	0.042	0.042	0.043	0.043	0.043	0.045	0.043
	Predict Date	31	32	33	34	35	36	37	38	39	40
	HDD	0.113	0.095	0.097	0.104	0.107	0.110	0.111	0.106	0.102	0.100
	NHD	0.041	0.041	0.043	0.046	0.050	0.045	0.045	0.043	0.046	0.049

**Table 7 sensors-25-03079-t007:** Experimental results for the sensitivity of the adaptive sampling function with respect to r.

*r*		0.1	0.2	0.3	0.4	0.5 (Ours)	0.8	1.0
1	HHD	0.0436	0.0372	0.0308	0.0281	0.063	0.0349	0.0462
	NHD	0.0392	0.0327	0.0274	0.0251	0.0241	0.0318	0.0415
2	HHD	0.0738	0.0631	0.0524	0.0489	0.0607	0.0794	0.0452
	NHD	0.0504	0.0421	0.0357	0.0332	0.0309	0.0419	0.0543
3	HHD	0.0835	0.0702	0.0589	0.0557	0.0503	0.0683	0.0881
	NHD	0.0581	0.0489	0.0413	0.0385	0.0355	0.0476	0.0612
4	HHD	0.0921	0.0794	0.0662	0.0618	0.0057	0.0765	0.0973
	NHD	0.0579	0.0489	0.0411	0.0389	0.0351	0.0474	0.0611
5	HHD	0.0954	0.0819	0.0687	0.0632	0.0569	0.0789	0.1012
	NHD	0.0653	0.0552	0.0468	0.0437	0.0399	0.0539	0.0691
6	HHD	0.0987	0.0853	0.0715	0.0651	0.0584	0.0814	0.1056
	NHD	0.0682	0.0578	0.0491	0.0459	0.0414	0.0564	0.0723
7	HHD	0.1002	0.0867	0.0728	0.0664	0.0588	0.0829	0.1078
	NHD	0.0707	0.0599	0.0511	0.0477	0.0431	0.0585	0.0751
8	HHD	0.1073	0.0921	0.0775	0.0719	0.0639	0.0883	0.1154
	NHD	0.0724	0.0614	0.0523	0.0489	0.0444	0.0599	0.0768
9	HHD	0.1146	0.0984	0.0829	0.0772	0.0689	0.0947	0.1231
	NHD	0.0779	0.0661	0.0564	0.0528	0.0477	0.0643	0.0821
10	HHD	0.1198	0.1037	0.0873	0.0815	0.0715	0.0992	0.1289
	NHD	0.0815	0.0693	0.0592	0.0554	0.0672	0.0857	0.0511

## Data Availability

The data presented in this study are available on request from the corresponding author.

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
