# Peer review of "Natural Gas Consumption Forecasting Model Based on Feature Optimization and Incremental Long Short-Term Memory"

_sensors, 2025, doi:10.3390/s25103079_

Round 1

Reviewer 1 Report

Comments and Suggestions for Authors

The paper presents a approach to natural gas consumption forecasting with promising results. However, methodological clarity and rigorous validation need minor improvements. 

In line 66 : The explanation of the adaptive function (Eq. 4) is unclear. How the Gaussian Mixture Model is applied to missing/anomalous data? Could you, provide a sensitivity analysis for r to justify its selection?

Please explain measures taken to prevent overfitting (e.g., early stopping, validation splits). Could you provide training/validation loss curves in supplementary materials?

Could you, Include short comparisons with modern models (e.g., Transformers, Temporal Fusion Transformers) to better establish the proposed method’s superiority over existing approaches?

There are some errors and inaccuracies:

- in line 217 the symbol in the formula is displayed incorrectly;

- in line 286 Error with reference;

- the abbreviation BP was not deciphered before its first mention. 

Author Response

Thanks for your suggestion. We have revised the paper according to your suggestion, and the response to each comment can be found in the following attchment.

Reviewer 2 Report

Comments and Suggestions for Authors

The paper presents an innovative approach to natural gas consumption forecasting model based on feature optimization and incremental LSTM. The presented advanced techniques were validated by forecasting natural gas load for Wuhan from 2011 to 2024. The use of LSTM networks for time series prediction models demonstrates a practical application in domain of energy forecasting.

The article's methodology is clearly explained, and the steps taken are outlined in detail.

Although the hypothesis is clear (the proposed model improves forecast accuracy), the work could benefit from testing on more diverse data sets (different regions, different time periods) to further validate the model in a general way.

The data and analysis are mostly clearly shown, but some figures and explanations could be improved for more clarity (figures 4-11).

The paper uses standard evaluation metrics (MAPE) and presents results in a way that allows for comparison with other methods.

The bibliographical references are perfectly aligned with each stage of the article, enhancing its success.

My minor comments are:

Line 217 – Clarify the expression: The input data xt

Line 254-255 - Eq 12 – Insert the superior limit on the sum operator

Line 286 - Please remove textError! Reference source not found” and insert the right reference

Line 341 – Insert the right reference in text “Observations from Figure 10 and Figure 10”

Line 359 – The item number is wrong (4 is right) -> 3) Prediction comparison of weather features with different durations:

Suggestion for authors - a paragraph in which the model presented in the article can be applied to other fields (renewable energy forecasting, smart grid management, etc.).

I recommend that the authors insert a nomenclature with acronyms, abbreviations and symbols used in the paper (used to improve clarity for readers).

Overall, I think the article is good enough to be approved, after minor revisions.

Author Response

(The authors gave the same response as above.)

Reviewer 3 Report

Comments and Suggestions for Authors

The paper is devoted to the development of a natural gas consumption forecasting model based on feature optimization and LSTM incremental learning. The authors propose a new approach to improve the accuracy of forecasts, taking into account various factors affecting gas consumption. The relevance of the work is determined by the growing need for accurate energy consumption forecasting, which is necessary to ensure reliable energy supplies, plan energy infrastructure and optimize resource use. In the context of volatile energy markets and a changing climate, accurate forecasts are becoming critical for making informed decisions in the energy sector. The importance of the work lies in the innovative methods proposed by the authors. The use of Gaussian mixtures to handle missing and anomalous data, the development of a weakly supervised cascade network for feature selection, and the implementation of incremental learning and regression difference loss function - all this is aimed at improving the accuracy and reliability of forecasts. The results obtained on gas consumption data in Wuhan demonstrate the superiority of the proposed method over traditional approaches. This work can make a significant contribution to the development of energy consumption forecasting methods and has practical implications for energy companies and government organizations.
There are several comments on the article that need to be improved.
1. Add keywords up to 7-8. Expand the abstract a little by adding more numerical results obtained in the work.
2. The beginning of the article contains too general statements about the importance of natural gas. It is necessary to more specifically define the problem of consumption forecasting and its impact on the variations in natural gas consumption will be improved by the proposed method, and what specific indicators will be used to evaluate the effectiveness of the model.
3. Section "Materials and Methods". It is necessary to indicate the sources of data, the period of time for which the data were collected, and also describe in more detail the characteristics of each feature. It is also important to discuss possible limitations associated with the use of a specific dataset and how they may affect the generalizability of the results.
4. The description of incremental learning is not detailed enough. It is necessary to explain how exactly the model adapts to new training and test samples, and what metrics were used to evaluate the effectiveness of the model.
5. The results of the comparison of the proposed method with the BP neural network do not look convincing enough. It is necessary to compare the proposed method with modern state-of-the-art (SOTA) approaches to forecasting natural gas consumption to demonstrate its competitiveness.
6. It is necessary to conduct a more detailed analysis of forecasting errors. It is important to identify the reasons why the model makes errors in certain cases and propose possible ways to eliminate them. It is also necessary to consider which factors have the greatest impact on the accuracy of forecasts.
7. The conclusions made in the article seem somewhat optimistic, given the limitations of the presented results. It is necessary to formulate conclusions more carefully and acknowledge the possible limitations of the proposed method.
8. The presented results are obtained using gas consumption data in Wuhan. It is necessary to discuss how generalizable these results are to other regions and countries. It is also important to consider which factors may affect the effectiveness of the proposed method in other conditions.

Author Response

(The authors gave the same response as above.)

Round 2

Reviewer 3 Report

Comments and Suggestions for Authors

The authors have thoroughly revised the article based on my comments. A small revision could clarify the methodology of comparison with SOTA (model settings, data sets, criteria for equality of conditions). Also add a practical example of an error and the model's reaction to it.

Author Response

Thanks for your suggestion. We have made revision according to your suggestion. The detailed response can be found in the following attachment.
